# Complemental Diagnosis of IgG4-Related Pancreaticobiliary Diseases by Multiple Hypoechoic Lesions in the Submandibular Glands

**DOI:** 10.3390/jcm11144189

**Published:** 2022-07-19

**Authors:** Naruki Shimamura, Shinichi Takano, Mitsuharu Fukasawa, Makoto Kadokura, Hiroko Shindo, Ei Takahashi, Sumio Hirose, Yoshimitsu Fukasawa, Satoshi Kawakami, Hiroshi Hayakawa, Natsuhiko Kuratomi, Hiroyuki Hasegawa, Shota Harai, Dai Yoshimura, Naoto Imagawa, Tatsuya Yamaguchi, Taisuke Inoue, Shinya Maekawa, Tadashi Sato, Nobuyuki Enomoto

**Affiliations:** 1First Department of Internal Medicine, Faculty of Medicine, University of Yamanashi, 1110, Shimokato, Chuo 409-3898, Yamanashi, Japan; nshimamura@yamanashi.ac.jp (N.S.); fmitsu@yamanashi.ac.jp (M.F.); m.k12095@kmh.kofu.yamanashi.jp (M.K.); hillos0202@gmail.com (H.S.); etakahashi@yamanashi.ac.jp (E.T.); sh99073@yahoo.co.jp (S.H.); mfukasawa@yamanashi.ac.jp (Y.F.); skawakami@yamanashi.ac.jp (S.K.); hhayakawa@yamanashi.ac.jp (H.H.); nkyamanashi@gmail.com (N.K.); hirohasegawa@yamanashi.ac.jp (H.H.); sharai@yamanashi.ac.jp (S.H.); dyoshimura@yamanashi.ac.jp (D.Y.); nimagawa@yamanashi.ac.jp (N.I.); ytatsuya@yamanashi.ac.jp (T.Y.); tinoue@yamanashi.ac.jp (T.I.); maekawa@yamanashi.ac.jp (S.M.); satou-tadashi@yamanashi.jcho.go.jp (T.S.); enomoto@yamanashi.ac.jp (N.E.); 2Department of Gastroenterology, Kofu Municipal Hospital, 366, Masutsubo, Kofu 400-0832, Yamanashi, Japan; 3Department of Gastroenterology, Nirasaki Municipal Hospital, 3-5-3, Honmachi, Nirasaki 407-0024, Yamanashi, Japan; 4Department of Gastroenterology, Yamanashi Prefectural Central Hospital, 1-1-1, Fujimi, Kofu 400-8506, Yamanashi, Japan; 5Department of Gastroenterology, Yamanashi Kosei Hospital, 860, Ochiai, Yamanashi 405-0033, Yamanashi, Japan; 6Department of Gastroenterology, Yamanashi Hospital, 3-11-16, Asahi, Kofu 400-0025, Yamanashi, Japan

**Keywords:** autoimmune pancreatitis, IgG4-SC, submandibular glands, sialadenitis

## Abstract

The diagnosis of autoimmune pancreatitis (AIP) and immunoglobulin G4-related sclerosing cholangitis (IgG4-SC) may require a somewhat invasive pathological examination and steroid responsiveness. This retrospective study assessed the complemental diagnosis of AIP and IgG4-SC using submandibular gland (SG) ultrasonography (US) in 69 patients, including 54 patients with AIP, 2 patients with IgG4-SC, and 13 patients with both AIP and IgG4-SC. The data from the physical examination and US of SGs to diagnose AIP (*n* = 67) and IgG4-SC (*n* = 15) were analyzed. The steroid therapy efficacy in resolving hypoechoic lesions in SGs was evaluated in 36 cases. The presence of IgG4-related pancreaticobiliary disease with multiple hypoechoic lesions in SGs was reduced from 31 to 11 cases after steroid therapy, suggesting that multiple hypoechoic lesions in SGs are strongly associated with IgG4-positive cell infiltrations. Multiple hypoechoic lesions in SGs were observed in 53 cases, whereas submandibular swelling on palpation was observed in 21 cases of IgG4-related pancreaticobiliary diseases. A complemental diagnosis of IgG4-related pancreaticobiliary diseases without a histological diagnosis and steroid therapy was achieved in 57 and 68 cases without and with multiple hypoechoic lesions in SGs, respectively. In conclusion, multiple hypoechoic lesions in SGs are useful for the complemental diagnosis of IgG4-related pancreaticobiliary diseases.

## 1. Introduction

Immunoglobulin G4-related disease (IgG4-RD) was recently established as a disease concept and is now widely recognized [1,2,3]. IgG4-RD includes pancreaticobiliary manifestations, such as autoimmune pancreatitis (AIP) and IgG4-related sclerosing cholangitis (IgG4-SC). AIP is a distinct type of pancreatitis characterized by periductal inflammation [4]. The typical radiological and clinical findings of AIP include diffuse or segmental narrowing of the main pancreatic duct, swelling of the pancreas, elevated serum levels of immunoglobulin G4 (IgG4), extrapancreatic lesions with abundant infiltration of IgG4-positive plasma cells, and responsiveness to corticosteroid therapy [4,5]. In 2011, the International Consensus Diagnostic Criteria (ICDC) were implemented for the systematic diagnosis of AIP, based on the following five cardinal features: serology, pancreatic parenchyma and duct imaging, other organ involvement (OOI), histology of the pancreas, and response to steroid therapy. Each feature is categorized as level 1 or 2 according to its specificity and reliability [4]. IgG4-SC is a distinct type of cholangitis that is frequently associated with AIP and is recognized as part of IgG4-RD. The clinical diagnostic criteria for IgG4-SC were established in 2012 and include biliary tract imaging, serum IgG4 levels, OOI, and histopathology of the biliary epithelium [6,7]. According to these diagnostic criteria, typical cases are now easily diagnosed; however, the response to steroid therapy, which can cause adverse events, and invasive histological examinations, such as endoscopic ultrasound with fine-needle biopsy and endoscopic bile duct biopsy, are needed for diagnosing atypical cases of AIP and IgG4-SC.

AIP and IgG4-SC frequently involve diseases in other organs, such as sclerosing sialadenitis, sclerosing dacryoadenitis, retroperitoneal fibrosis, and pseudotumors. Information regarding the involvement of other organs increases diagnostic sensitivity. The histology of these lesions is similar to that of the pancreas in patients with AIP and includes an abundant infiltration of IgG4-positive plasma cells and lymphocytes with dense fibrosis [8,9]. Sclerosing sialadenitis is a feature of Mikulicz’s disease manifesting as bilateral, painless, and symmetrical swelling of the parotid and submandibular glands (SGs) [10]. Incidentally, SGs are located near the body surface and are easy to evaluate using ultrasonography (US). However, the ultrasonographic characteristics of SGs have only been documented in a small group of patients with AIP (*n* = 9) [11] and the diagnostic criteria for sialadenitis are not well-documented in the ICDC.

We investigated the hypoechoic nodules of SGs in patients with AIP and performed SG US in patients with IgG4-related disease at our hospital. Moreover, we previously reported that multiple hypoechoic lesions in SGs are a specific marker of AIP that can be objectively and noninvasively diagnosed by US [12,13]. However, neither the significance of these lesions for the diagnosis of sialadenitis nor their implications in IgG4-related pancreaticobiliary diseases are known. In this study, we aimed to determine the significance of hypoechoic SG lesions in the diagnosis of AIP and IgG4-SC using US.

## 2. Materials and Methods

### 2.1. Patients and Diagnosis

The medical records of 69 patients, including 54 patients diagnosed with AIP, 2 patients diagnosed with IgG4-SC, and 13 patients diagnosed with both AIP and IgG4-SC at our institution between October 2003 and October 2020 were retrospectively reviewed. Therefore, the cohort included 67 AIP cases and 15 IgG4-SC cases. AIP was diagnosed according to the Japanese clinical diagnostic criteria [14] or ICDC [4], and IgG4-SC was diagnosed based on the clinical diagnostic criteria [15]. Diagnoses of extrapancreatic/extrabiliary organ involvement were established according to the comprehensive diagnostic criteria for IgG4-related diseases and organ-specific diagnostic criteria [2]. Sialadenitis was diagnosed based on the presence of a bilateral symmetrical enlargement, according to the AIP diagnostic criteria [14]. Although not all AIP cases met the diagnostic criteria for a definitive diagnosis, the patients who met at least a suspected diagnosis according to either the Japanese diagnostic criteria or the ICDC and whose clinical courses were consistent with AIP were included in this study. These criteria were designated as the gold standards. All patients were evaluated by SG US at least once, and 35 patients with AIP and 8 patients with IgG4-SC were evaluated using SG US before and after steroid therapy. This retrospective study was approved by the Ethics Committee of Yamanashi University Hospital (receipt number: 2486), which waived the requirement for written informed consent because the study was a retrospective data analysis, with appropriate consideration given to patient risk, privacy, welfare, and rights.

### 2.2. Ultrasonographic Examination of Submandibular Glands

The ultrasonographic examination of SGs was performed using a ProSound α10^®^ instrument (Aloka, Tokyo, Japan) at a frequency of 7.5 MHz. The thickest part of each SG was measured, and the characteristic features were analyzed by two US experts to reach a diagnosis. The SG findings were classified into homogeneous and multiple hypoechoic lesions based on the US (Figure 1), as reported previously [12,13]. The differential diagnosis of hypoechoic lesions in SGs included Sjogren syndrome and lymphoma, which were ruled out by measuring anti-SS-A and anti-SS-B antibody levels (markers of Sjogren syndrome) and the clinical course of the disease.

### 2.3. Statistical Analysis

The quantitative data were expressed as means ± standard deviation, and the continuous data were expressed as medians and interquartile ranges (IQRs). Two-group comparisons were conducted using paired *t*-tests or chi-squared tests. The significance was set at *p* values < 0.05.

## 3. Results

### 3.1. Patient Characteristics and Classification of US Findings in Submandibular Glands

Table 1 lists the clinical characteristics of all patients included in the present study. The patients diagnosed with AIP (*n* = 67) and IgG4-SC (*n* = 15) were analyzed. The SGs were evaluated by ultrasound in all cases. Based on US findings, the SGs were classified according to our previous reports into the following three groups: homogeneous, heterogeneous, and multiple hypoechoic lesions [13]; multiple hypoechoic lesions were often observed in patients with AIP (Figure 1a–d).

### 3.2. Comparison of US Findings in Submandibular Glands before and after Steroid Therapy

To elucidate the pathophysiology and significance of multiple hypoechoic lesions in SGs, we compared the US findings of SGs before and after steroid therapy. The US findings of the SGs in 35 patients with AIP and 8 patients with IgG4-SC were evaluated before and after steroid therapy. Table 2 reports the changes in US findings in the SGs before and after steroid therapy. Multiple hypoechoic lesions in the SGs were observed before steroid therapy in 31 (88%) patients with AIP, 7 (87%) patients with IgG4-SC, and 31 (86%) cases in total. After steroid therapy, the number of cases with multiple hypoechoic lesions in the SGs was significantly reduced to 11 (31%, *p* < 0.001), 2 (25%, *p* = 0.039), and 11 (31%, *p* < 0.001) cases with AIP, IgG4-SC, and both AIP and IgG4-SC, respectively (Table 2), implying that multiple hypoechoic lesions in SGs correspond to the accumulation of IgG4-positive lymphocytes. The representative US changes in the SGs observed before and after steroid therapy are shown for one case in Figure 1e,f; the hypoechoic lesions were resolved after steroid therapy in this case.

### 3.3. Sensitivity for Diagnosing Sialadenitis in AIP and IgG4-SC

To clarify the usefulness of multiple hypoechoic lesions in the diagnosis of sialadenitis in patients with AIP and IgG4-SC, comparisons with other tests, including submandibular swelling on palpation and SG thickness on US, were performed (Table 3). Among the patients with AIP, 52 (78%) cases were positive for multiple hypoechoic lesions. In turn, significantly fewer cases were positive when using other tests; 21 (31%, *p* = 0.019) and 28 (42%, *p* = 0.052) cases were positive using palpation of the SGs and SG thickness on US, respectively. Among the patients with IgG4-SC, multiple hypoechoic lesions were detected in 10 (67%) cases, whereas 3 (20%, *p* = 0.73) and 6 (40%, *p* = 0.29) cases were positive when using palpation of the SGs and SG thickness on US, respectively. In all cases, multiple hypoechoic lesions were detected in 53 (77%) cases, whereas 21 (30%, *p* = 0.016) and 29 (42%, *p* = 0.031) cases were positive when using palpation of the SGs and SG thickness on US, respectively.

### 3.4. Multiple Hypoechoic Lesions Facilitate the Complemental Diagnosis of IgG4-Related Pancreaticobiliary Disease

The comparison of the diagnosis of AIP or IgG4-SC using invasive examinations, such as the pathological examination and the effects of steroid therapy, with the diagnosis of sialadenitis using multiple hypoechoic lesions is shown in Figure 2a–c. AIP was noninvasively diagnosed in 66% and 81% of the 67 cases without and with the use of multiple hypoechoic lesions in SGs, respectively (*p* = 0.051). A noninvasive diagnosis of IgG4-SC was established in 87% and 93% of 15 cases without and with the use of multiple hypoechoic lesions in SGs, respectively (*p* = 0.540). A noninvasive diagnosis was established in all IgG4-related pancreaticobiliary diseases (combined AIP and IgG4-SC) in 65% and 81% of 69 cases without and with the use of multiple hypoechoic lesions of the SGs, respectively (*p* = 0.043).

### 3.5. A Representative Case in Which US Findings of SGs Might Be Useful

A list of cases that show the effect of the US findings of SGs in diagnosing AIP is provided in Table 4. There were 51 cases of definitively diagnosed AIP among all 67 cases, which also included possible or probable cases if US findings of SGs were not used for diagnosing AIP because the SG findings were not introduced to the diagnostic criteria. EUS-FNA and a steroid trial were exempted in 1 and 5 cases among the definitively diagnosed 51 cases, respectively, and they were also exempted in 1 and 9 cases, respectively, among all 67 cases with AIP, when the US findings of SGs were introduced to the diagnostic criteria. Additionally, the diagnosis would be upgraded in 8 of 67 cases with AIP, highlighting the complemental diagnostic ability of SG US findings.

In a representative case, AIP was definitively diagnosed using the detection of the presence of sialadenitis based on multiple hypoechoic lesions (Figure 3a–f) before confirming the diagnosis based on steroid responsiveness. The case was a woman in her 60s. Computed tomography (CT) showed the focal enlargement of the pancreas (Figure 3a). Endoscopic retrograde pancreatography (ERP) revealed an irregular narrowing of the main pancreatic duct without a marked upstream pancreatic duct dilation (Figure 3c). The serum IgG4 level was 217 mg/dL, and no extrapancreatic lesions were observed. Therefore, the probable diagnosis in this patient, according to the Japanese clinical diagnostic criteria, was AIP. The pathological examination showed no characteristic findings of AIP, and no findings suggestive of pancreatic cancer were observed. Thus, the effectiveness of steroid therapy was examined.

After steroid therapy, the focal enlargement of the pancreas on CT was less noticeable (Figure 3b) and the irregular narrowing of the main pancreatic duct on ERP was also improved (Figure 3d). Thus, steroid therapy was effective, and we were able to establish a definitive diagnosis. The ultrasonographic examination of the SGs in this case showed multiple hypoechoic lesions before steroid therapy (Figure 3e), which were obscured after steroid therapy (Figure 3f). If a diagnosis of sialadenitis had been made based on multiple hypoechoic lesions, the presence of extrapancreatic lesions would have been observed and a definitive diagnosis could have been established before the effectiveness of the steroid therapy was observed.

## 4. Discussion

This study demonstrated that the identification of sialadenitis based on the US findings of multiple hypoechoic lesions in SGs may help avoid the use of invasive examinations, such as endoscopic ultrasound with fine-needle biopsy and steroid therapy, for the diagnosis of IgG4-related pancreaticobiliary diseases, including AIP and IgG4-SC. Thus, the noninvasive and objective diagnosis of multiple hypoechoic lesions in SGs using US might have wide utility for the complemental diagnosis of IgG4-related pancreaticobiliary diseases, which warrants future prospective studies in sialadenitis and IgG4-related diseases.

The diagnosis of AIP and IgG4-SC was made easier after the establishment of the diagnostic criteria, such as the ICDC [4] and the Japanese Clinical Diagnostic Criteria for Autoimmune Pancreatitis in 2018 [14] for AIP and the clinical diagnostic criteria of IgG4-related sclerosing cholangitis in 2020 for IgG4-SC [15,16]. However, some cases of IgG4-RD are still difficult to definitively diagnose. These diagnostic criteria have been implemented for the systematic diagnosis of these complicated diseases based on the following five cardinal features: imaging of the pancreatic duct and/or parenchyma, serological findings, pathological findings, OOI, and the effectiveness of steroid therapy. Among these criteria, the effectiveness of steroid therapy should be used with caution because some pancreatic cancers and malignant lymphoma shrink after steroid treatment [4,17,18]. Furthermore, endoscopic ultrasound-guided fine-needle aspiration (EUS-FNA) for a histological diagnosis is associated with a 1.7% incidence of adverse events, including bleeding and pancreatitis [19]. In addition, according to a nationwide epidemiological survey of AIP in Japan in 2016, more than 60% of cases could not be histologically diagnosed by EUS-FNA alone [20]. According to an Italian epidemiological study, an AIP diagnosis was established in 75/173 (43%) patients based on the effects of steroid therapy, and 70/75 (93%) cases were diagnosed with AIP [21], although the diagnosis of this disease is premised on the exclusion of malignant tumors by pathological examination of the patients. Therefore, the diagnosis of AIP and IgG4-SC without the use of these two criteria is preferable, especially in older patients, who are likely to experience more severe adverse events. In the present study, we demonstrated that the use of ultrasonographic findings of multiple hypoechoic nodules in SGs reduced the dependence on invasive criteria, such as a histological diagnosis and the effectiveness of steroid therapy; sialadenitis could be diagnosed with high sensitivity based on characteristic and conspicuous ultrasonographic findings.

The ultrasonographic findings of multiple hypoechoic nodules in SGs contributed to the noninvasive and sensitive diagnosis of sialadenitis in IgG4-related pancreaticobiliary diseases. We previously demonstrated that hypoechoic nodules are useful in differentiating AIP from pancreatic cancer [12]. Furthermore, the ultrasonographic findings of hypoechoic nodules in sialadenitis can be easily, noninvasively, and objectively diagnosed by any physician [13]. These characteristic SG findings can be detected even in the absence of swelling of the salivary glands. Thus, ultrasound detection of hypoechoic nodules is a highly sensitive method of diagnosing sialadenitis in IgG4-related pancreaticobiliary diseases. Hypoechoic nodules observed in SGs are likely due to infiltrating IgG4-positive lymphocytes because the hypoechoic nodules disappeared or were obscured by steroid therapy in the current study. A previous retrospective study of US findings in the SGs of patients with IgG4-related sialadenitis showed that the region of lymphoplasmacytic infiltration corresponded to the hypoechoic region observed on the US [22].

The findings of the present study have several clinical implications. First, the characteristic findings of multiple hypoechoic nodules in the submandibular glands, which were present even in cases without palpation, can be an objective and sensitive diagnostic tool for IgG4-related sialadenitis. Second, the use of the ultrasonographic findings of multiple hypoechoic nodules in SGs reduced the number of cases requiring invasive testing for diagnosing IgG4-related pancreaticobiliary disease by 13%, as shown in Figure 2c. We recognize that EUS-FNA is important in AIP diagnosis in order to exclude malignant diseases, but the incidence of EUS-FNA-related adverse events is approximately 1.7% [19]. Furthermore, the rate of level 1 histology for AIP in ICDC is only 60.1% (95% CI: 42%–79%) [23]. We believe that a noninvasive test is useful for patients who are hesitant to undergo EUS-FNA due to patient background or comorbidities.

This study has several limitations inherent to its retrospective design. First, the number of cases included in the analysis, especially that of IgG4-SC alone (*n* = 2), was not sufficient, which made it impossible to obtain statistically significant differences in some analyses, including the IgG4-SC cases alone. Second, considering both the decrease in hypoechoic nodules after the steroid treatment and the previously reported findings, we indirectly demonstrated that the hypoechoic nodules were caused by inflammation triggered by IgG4-positive cell infiltration; however, we could not establish a direct relationship between the pathological and US findings. We previously reported no hypoechoic nodules in the control group of pancreatic cancer when we compared the SGUS findings in pancreatic cancer and AIP [12]; however, in future studies, needle biopsy or cytology testing should be performed to exclude other diseases.

## 5. Conclusions

The presence of multiple hypoechoic lesions in the SGs is a useful tool for the complemental diagnosis of IgG4-related pancreaticobiliary diseases without the use of steroids. The US findings also implied the presence of IgG4-positive cell infiltrations into the SGs. We hope that the findings of this study will contribute to the accurate diagnosis of IgG4-related pancreaticobiliary diseases without adverse effects on patients.

## Figures and Tables

**Figure 1 jcm-11-04189-f001:**
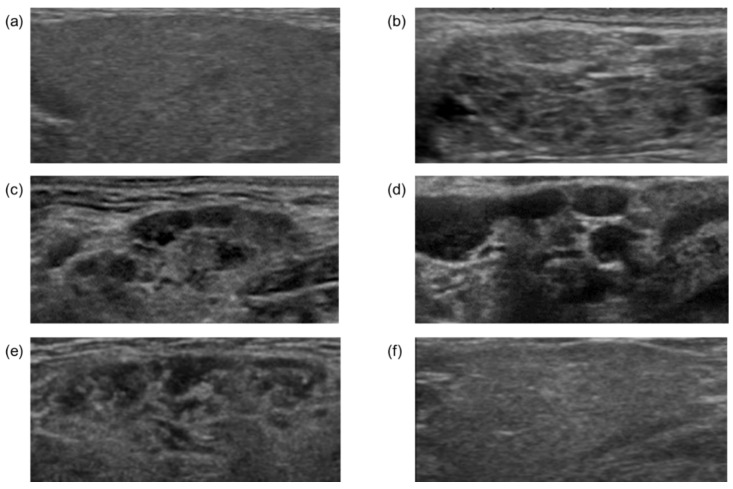
Representative ultrasonographic findings of submandibular glands. Homogeneous (**a**), heterogeneous (**b**), and multiple hypoechoic (**c**,**d**) nodules. Ultrasonographic findings of submandibular glands before (**e**) and after (**f**) steroid administration in the same case.

**Figure 2 jcm-11-04189-f002:**
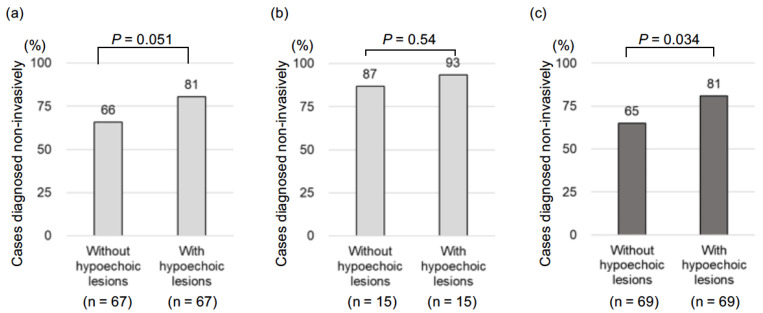
Percentages of cases that were diagnosed noninvasively (without pathological tests and steroid trials) in patients diagnosed with or without the use of ultrasonographic findings of multiple hypoechoic nodules in the submandibular glands. (**a**) Autoimmune pancreatitis (AIP), (**b**) IgG4-related sclerosing cholangitis (IgG4-SC), and (**c**) AIP and IgG4-SC.

**Figure 3 jcm-11-04189-f003:**
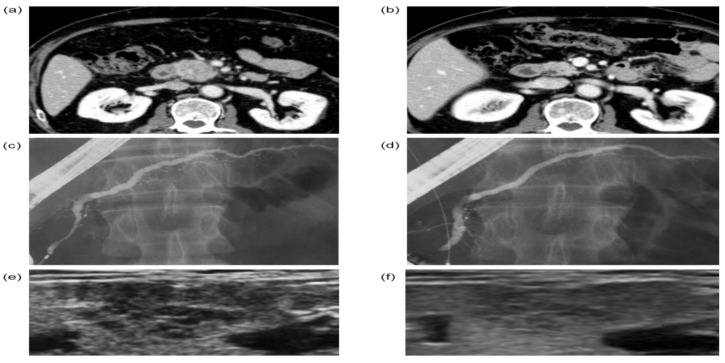
A representative case in which a definitive diagnosis of AIP was established using the presence of multiple hypoechoic nodules indicative of sialadenitis before a steroid trial. CT findings before (**a**) and after (**b**) the steroid trial. Endoscopic retrograde cholangiopancreatography findings before (**c**) and after (**d**) the steroid trial. Sialadenitis could not be diagnosed using palpation in this case; however, ultrasonographic findings of multiple hypoechoic nodules were observed before steroid therapy (**e**), which became obscured after steroid therapy (**f**).

**Table 1 jcm-11-04189-t001:** Patient characteristics.

	IgG4-SC	AIP
	(*n* = 15)	(*n* = 67)
Age (years), mean (± SD)	67.4 (±7.1)	66.1 (±9.3)
Sex, *n* (%)		
Male	11 (73)	45 (67)
Female	4 (27)	22 (33)
Serum IgG4 (mg/dL), median (IQR)	464 (292–680)	418 (296–697)
Extrapancreatic or extrabiliary lesions		
AIP	13 (87)	N/A
IgG4-SC	N/A	13 (19)
Dacryoadenitis/sialadenitis	7 (47)	42 (63)
Retroperitoneal fibrosis	1 (7)	9 (13)
Diagnosis by Japanese clinical diagnostic criteria, *n* (%)		
Definite	14 (93)	50 (75)
Probable or possible	1 (7)	17 (25)
Diagnosis by ICDC, *n* (%)		
Definite	N/A	59 (88)
Probable	N/A	6 (9)
Not met	N/A	2 (3)

IgG4-SC, IgG4-related sclerosing cholangitis; AIP, autoimmune pancreatitis; IQR, interquartile range; N/A, not applicable; ICDC, International Consensus Diagnostic Criteria.

**Table 2 jcm-11-04189-t002:** Ultrasonography findings of the submandibular glands before and after steroid therapy.

	Before	After	*p* *
US findings of the SG	steroid therapy	steroid therapy	
IgG4-SC (*n* = 8)			0.012
Homogeneous, *n* (%)	0 (0%)	1 (13%)	
Heterogeneous ^†^, *n* (%)	1 (13%)	6 (75%)	
Multiple hypoechoic lesions, *n* (%)	7 (87%)	1 (13%)	
AIP (*n* = 35)			<0.001
Homogeneous, *n* (%)	2 (6%)	4 (11%)	
Heterogeneous ^†^, *n* (%)	2 (6%)	28 (80%)	
Multiple hypoechoic lesions, *n* (%)	31 (88%)	3 (9%)	
All (*n* = 36)			<0.001
Homogeneous, *n* (%)	2 (6%)	4 (11%)	
Heterogeneous ^†^, *n* (%)	3 (8%)	29 (81%)	
Multiple hypoechoic lesions, *n* (%)	31 (86%)	31 (8%)	

US, ultrasonography; IgG4-SC, IgG4-related sclerosing cholangitis; AIP, autoimmune pancreatitis. * *p* value analyzed separately for homogeneous and heterogeneous cases. ^†^ Heterogenous includes cases with reduced hypoechoic lesions in SGs by steroid treatment.

**Table 3 jcm-11-04189-t003:** Prevalence of sialadenitis diagnosed based on the presence of swelling or hypoechoic lesions in patients with AIP and IgG4-SC.

		Swelling on Palpation	US Findings of Submandibular Glands		*p* ^‡^	*p* ^§^
	Total Cases		Thickness *	Hypoechoic Lesions ^†^		
IgG4-SC, *n* (%)	15	3 (20)	6 (40)	10 (67)	0.730	0.290
AIP, *n* (%)	67	21 (31)	28 (42)	52 (78)	0.019	0.052
All, *n* (%)	69	21 (30)	29 (42)	53 (77)	0.016	0.031

US, ultrasonography; IgG4-SC, IgG4-related sclerosing cholangitis; AIP, autoimmune pancreatitis. * thickness of the submandibular gland measured by US; ^†^ US findings of multiple hypoechoic lesions; ^‡^
*p* value for the comparison of the sensitivity of hypoechoic nodule findings with the sensitivity of palpation; ^§^
*p* value for the comparison of the sensitivity of hypoechoic nodule findings with the sensitivity of SG thickness on echo.

**Table 4 jcm-11-04189-t004:** A list of cases that shows the effect of US findings of SGs in diagnosing AIP.

			Without SGUS		With SGUS		
Case	Age	Sex	Diagnosis *	FS ^†^	Diagnosis *	FS ^†^	Diagnostic Effectiveness of SGUS
1	72	M	Definite: IVa	F	Definite: Ib + IIa < III/IVb/V(a/b)> ^‡^		EUS-FNA exemption
2	63	F	Definite: Ib + II < III/IVb/V(a/b)VI	S	Definite: Ib + IIa < III/IVb/V(a/b)> ^‡^		Steroid trial exemption
3	57	F	Definite: Ib + II < III/IVb/V(a/b) > VI	S	Definite: Ib + IIa < III/IVb/V(a/b)> ^‡^		Steroid trial exemption
4	67	M	Definite: Ib + II < III/IVb/V(a/b) > VI	S	Definite: Ib + IIa < III/IVb/V(a/b)> ^‡^		Steroid trial exemption
5	58	M	Definite: Ib + II < III/IVb/V(a/b) > VI	S	Definite: Ib + IIa < III/IVb/V(a/b)> ^‡^		Steroid trial exemption
6	67	F	Definite: Ib + II < III/IVb/V(a/b) > VI	S	Definite: Ib + IIa < III/IVb/V(a/b)> ^‡^		Steroid trial exemption
7	64	F	Possible: Ia + II + VI	S	Definite: Ia < III/IVb/V(a/b)>	S	Upgrade
8	62	F	Probable: Ib + IIa < III/IVb/V(a/b)>		Definite: Ib + IIa < III/IVb/V(a/b)> ^‡^		Upgrade
9	52	M	Probable: Ib + IIa < III/IVb/V(a/b)>		Definite: Ib + IIa < III/IVb/V(a/b)> ^‡^		Upgrade
10	79	F	Probable: Ib + IIa < III/IVb/V(a/b)>		Definite: Ib + IIa < III/IVb/V(a/b)> ^‡^		Upgrade
11	75	M	Possible: Ib + II + VI	S	Probable: Ib + IIa < III/IVb/V(a/b)>		Upgrade and steroid trial exemption
12	52	M	Possible: Ib + II + VI	S	Probable: Ib + IIa < III/IVb/V(a/b)>		Upgrade and steroid trial exemption
13	57	M	Possible: Ib + II + VI	S	Probable: Ib + IIa < III/IVb/V(a/b)>		Upgrade and steroid trial exemption
14	40	F	Possible: Ib + IIa + VI	S	Probable: Ib + IIa < III/IVb/V(a/b)>		Upgrade and steroid trial exemption

* Diagnosis of autoimmune pancreatitis by Japanese clinical diagnostic criteria. ^†^ EUS-FNA or steroid trial needed for diagnosis. F, EUS-FNA; S, steroid trial. ^‡^ Two or more findings among <III/IVb/V(a/b)> are needed. SGUS, submandibular glands ultrasonography; <III/IVb/V(a/b)>, a positive finding among III, IVb, Va, or Vb; Ia, diffuse enlargement of the pancreas; Ib, segmental enlargement of the pancreas; II, irregular narrowing of the main pancreatic duct by ERP (IIa) or MRCP (IIb); III, elevated serum IgG4; IV, pathological findings; V, extrapancreatic lesions including sclerosing cholangitis, sclerosing dacryoadenitis, sialadenitis, retroperitoneal fibrosis, and kidney lesions with clinical (Va) and pathological (Vb) lesions; VI, effectiveness of steroid therapy.

## Data Availability

Data sharing is not applicable for this article.

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
