# Peer review of "Complemental Diagnosis of IgG4-Related Pancreaticobiliary Diseases by Multiple Hypoechoic Lesions in the Submandibular Glands"

_jcm, 2022, doi:10.3390/jcm11144189_

Round 1

Reviewer 1 Report

Manuscript has been sufficiently improved

Reviewer 2 Report

All the queries raised have been satisfactorily answered. Only minor spell checks are required. Kindly do the needful.

This manuscript is a resubmission of an earlier submission. The following is a list of the peer review reports and author responses from that submission.

Round 1

Reviewer 1 Report

I was pleased to review the article   Non-invasive diagnosis of IgG4-related pancreatobiliary diseases by multiple hypoechoic lesions in the submandibular glands

This work is important, with an interesting and useful research for diagnosis of autoimmune pancreatitis and immunoglobulin G4-related sclerosing cholangitis.

The findings of this study may contribute to the accurate diagnosis of IgG4-related pancreatobiliary diseases without adverse effects on patients.

The manuscript is clear, relevant for the field. The methodology used by the authors is appropriate for the purpose of the study and conclusions are narrowly linked to data discussion and available evidence.

In general, the Manuscript may benefit from some minor revisions, as suggested below:

  • I suggest to respect the classic structure of the abstract- IMRAD
  • In the manuscript I received for review, there is no complete data for Fig. 1 , the corresponding explanations for the images are missing (a-f), and the same situation for fig. 2 and 3

Author Response

Responses to the Comments from Reviewer #1

We are grateful to Reviewer #1 for their critical comments and insightful suggestions, which were instrumental in improving our manuscript considerably. We have addressed all comments and have incorporated the necessary changes in the revised manuscript. Our point-by-point responses to the comments are provided below.

Comments to the Author from Reviewer #1

Comments to the Author

I was pleased to review the article “Non-invasive diagnosis of IgG4-related pancreatobiliary diseases by multiple hypoechoic lesions in the submandibular glands.” This work is important, with an interesting and useful research for diagnosis of autoimmune pancreatitis and immunoglobulin G4-related sclerosing cholangitis. The findings of this study may contribute to the accurate diagnosis of IgG4-related pancreatobiliary diseases without adverse effects on patients. The manuscript is clear, relevant for the field. The methodology used by the authors is appropriate for the purpose of the study and conclusions are narrowly linked to data discussion and available evidence. In general, the Manuscript may benefit from some minor revisions, as suggested below:

Comments

Q1. I suggest to respect the classic structure of the abstract- IMRAD

A1. We thank the Reviewer for their recommendation. We have added the following sentence in the Abstract to maintain the IMRAD structure. We have also revised the entire abstract to keep it within the 200-word limit.

“The diagnosis of autoimmune pancreatitis (AIP) and immunoglobulin G4-related sclerosing cholangitis (IgG4-SC) may require somewhat invasive pathological examination and steroid responsiveness.”

Q2. In the manuscript I received for review, there is no complete data for Fig. 1, the corresponding explanations for the images are missing (a-f), and the same situation for fig. 2 and 3

A2. We apologize for the incorrect placement of the Figure legends in our original manuscript. The legends have been placed directly below each figure in the revised manuscript.

Reviewer 2 Report

1)      On page 3, line 38, it is mentioned that 69 cases were included, but on the same page, line 47 mentions 82 cases. Kindly clarify the distribution of cases.

2)      Why only 35 and 8 patients of AIP and IgG4-SC (out of 69) were evaluated with SG US? Why rest of the patients were not considered for SG US?

3)      On page 6, line 104, word “was” is not required.

4)      How was the diagnosis of extra-pancreatic/extra-biliary organ involvement (dacryoadenitis/sialadenitis/retroperitoneal fibrosis) made?

5)      There were 50 definite cases of AIP by Japanese criteria and 59 definite cases of AIP by ICDC. Which one were considered as definite cases?

6)      Were the other causes of similar submandibular gland lesions ruled out?

7)      There is significant increase in heterogenous echogenicity SGs after steroid treatment. What is the signficance of this finding?

8)      The sensitivity of non-invasive diagnostic tests to diagnose sialadenitis was compared against which gold standard diagnostic test (? Biopsy)/criteria, as it is mentioned in the manuscript that diagnostic criteria for sialadenitis are not well documented in ICDC.

9)      Did all the patients enrolled in the study (who were diagnosed with IgG4 related pancreato-biliary disorder) receive steroid treatment? If they did receive steroids, then how SG US was recorded before steroid treatment, as all the patients were retrospectively enrolled.

10)   Kindly enumerate the limitations of the study.

11)   A prospective study design would be worthwhile before recommending that “US findings of multiple hypoechoic lesions in SGs, which can be diagnosed non-invasively and objectively, should be widely used for the diagnosis of IgG4-related pancreato-biliary diseases.”

12)   According to the study results, there is still 15-20 % risk of missing the diagnosis of IgG4-RD by using SG US, which seems significant, and the results of the study need to be confirmed with a bigger sample size before it can be uniformly considered for the diagnosis of IgG4-RD.

Author Response

Responses to the Comments from Reviewer #2

We would like to thank Reviewer #2 for their critical comments and useful suggestions, which have greatly enhanced the quality of our manuscript. As indicated in the point-by-point responses below, we have addressed all comments and have incorporated the corresponding changes in the revised manuscript.

Comments to the Author from Reviewer #2

Q1. On page 3, line 38, it is mentioned that 69 cases were included, but on the same page, line 47 mentions 82 cases. Kindly clarify the distribution of cases.

A1. We apologize for the confusion. Some of the patients had both AIP and IgG4-SC, which might have led to the confusion. Briefly, we examined the medical records of 69 patients, including 54 patients diagnosed with AIP, 2 patients diagnosed with IgG4-SC, and 13 patients diagnosed with both AIP and IgG4-SC. Therefore, the cohort included 67 AIP cases and 15 IgG4-SC cases.

We have therefore revised the sentence as follows (page 3, lines 36–40): “This retrospective study assessed the non-invasive diagnosis of AIP and IgG4-SC using submandibular gland (SG) ultrasonography (US) in 69 patients, including 54 patients with AIP, 2 patients with IgG4-SC, and 13 patients with both AIP and IgG4-SC. Data from physical examination and US of SGs to diagnose AIP (n = 67) and IgG4-SC (n = 15) were analyzed.”

In addition, we have revised the Methods section as follows (page 5, line 99 to page 6 line 102): “The medical records of 69 patients, including 54 patients diagnosed with AIP, 2 patients diagnosed with IgG4-SC, and 13 patients diagnosed with both AIP and IgG4-SC at our institution between October 2003 and October 2020 were retrospectively reviewed. Therefore, the cohort included 67 AIP cases and 15 IgG4-SC cases.”

Q2. Why only 35 and 8 patients of AIP and IgG4-SC (out of 69) were evaluated with SG US? Why rest of the patients were not considered for SG US?

A2. We apologize for the confusion. We evaluated 35 and 8 patients with AIP and IgG4-SC, respectively, using SG US both before and after steroid therapy. We performed SG US at least once in all patients.

We have revised the sentence by adding “both” to avoid confusion as follows (page 6, lines 112–114): “Thirty-five and eight patients with AIP and IgG4-SC, respectively, were evaluated by SG US both before and after steroid therapy.”

Q3. On page 6, line 104, word “was” is not required.

A3. We thank the Reviewer and apologize for the oversight. The sentence was revised as recommended in page 6, line 117 in the revised manuscript.

Q4. How was the diagnosis of extra-pancreatic/extra-biliary organ involvement (dacryoadenitis/sialadenitis/retroperitoneal fibrosis) made?

A4. The diagnosis of extrapancreatic/extrabiliary organ involvement was based on the comprehensive diagnostic criteria for IgG4-related diseases and organ-specific diagnostic criteria. Sialadenitis was diagnosed by bilateral symmetric enlargement according to the AIP diagnostic criteria.

We added the following to the revised manuscript (page 6, lines 104–108): “Diagnoses of extrapancreatic/extrabiliary organ involvement were established according to the comprehensive diagnostic criteria for IgG4-related diseases and organ-specific diagnostic criteria [2]. Sialadenitis was diagnosed based on the presence of bilateral symmetric enlargement, according to the AIP diagnostic criteria [14].”

We have also added the following citation for the diagnostic criteria: Kawa S, Kamisawa T, Notohara K, Fujinaga Y, Inoue D, Koyama T, et al. Japanese Clinical Diagnostic Criteria for Autoimmune Pancreatitis, 2018: Revision of Japanese Clinical Diagnostic Criteria for Autoimmune Pancreatitis, 2011. Pancreas.  

Q5. There were 50 definite cases of AIP by Japanese criteria and 59 definite cases of AIP by ICDC. Which one were considered as definite cases?

A5. Not all cases met the diagnostic criteria for the definitive diagnosis of AIP; however, those that met at least a suspected diagnosis based on either the Japanese diagnostic criteria or the ICDC and exhibited a clinical course that was consistent with AIP were included in the study. And these criteria were considered as the gold standard.

For clarity, we have added the following explanation in page 6, lines 108–112: “Although not all cases of AIP met the diagnostic criteria for a definitive diagnosis of AIP, the patients who met at least a suspected diagnosis according to either the Japanese diagnostic criteria or the ICDC and whose clinical course was consistent with AIP were included in this study. And these criteria were designated as the gold standard.”

Q6. Were the other causes of similar submandibular gland lesions ruled out?

A6. We thank the Reviewer for this important inquiry. The differential diagnosis of this condition includes Sjogren syndrome and lymphoma, which were ruled out by the measurement of anti-SS-A and anti-SS-B antibody levels, which are markers for Sjogren syndrome, as well as the clinical course of the disease.

We have added the following explanation (page 6, line 123 to page 7 line 127): “The differential diagnosis of hypoechoic lesions in SGs included Sjogren syndrome and lymphoma, which were ruled out by the measurement of anti-SS-A and anti-SS-B antibody levels (i.e., markers of Sjogren syndrome) and the clinical course of the disease.”

Q7. There is significant increase in heterogenous echogenicity SGs after steroid treatment. What is the significance of this finding?

A7. We assumed that the hypoechoic nodules detected in SGs corresponded to the infiltrating IgG4-positive lymphocytes, as described in page 19 (lines 286–289) and disappeared or were obscured after steroid treatment. Therefore, the heterogeneous echogenicity of SGs after steroid treatment may reflect a mechanism by which a hypoechoic lesion with a well-defined border is obscured by steroids.

Q8. The sensitivity of non-invasive diagnostic tests to diagnose sialadenitis was compared against which gold standard diagnostic test (? Biopsy)/criteria, as it is mentioned in the manuscript that diagnostic criteria for sialadenitis are not well documented in ICDC.

A8. We apologize for the confusion. The population parameter listed in Table 3 is not the number of sialadenitis cases; rather, it is the number of cases diagnosed with AIP and IgG4-SC, suggesting that the SG US findings may be able to diagnose sialadenitis, which cannot be diagnosed using the diagnostic criteria.

We have revised the title of Table 3 as follows: The prevalence of sialadenitis diagnosed based on the presence of swelling or hypoechoic lesions inpatients with AIP and IgG4-SC”

Q9. Did all the patients enrolled in the study (who were diagnosed with IgG4 related pancreato-biliary disorder) receive steroid treatment? If they did receive steroids, then how SG US was recorded before steroid treatment, as all the patients were retrospectively enrolled.

A9. In the present study, not all patients diagnosed with IgG4-related pancreaticobiliary disease were treated with steroids. We have long noted and reported the presence of hypoechoic nodules in the submandibular glands of patients with AIP and have performed SG US in principle in patients with IgG4-related disease in our hospital.

We have added the following explanation of the circumstances that led us to routinely perform SG US in patients with IgG4-related disease in our hospital (page 5, lines 89–91): “We have long been focusing on the hypoechoic nodules of the SGs observed in patients with AIP and have performed SG US in principle in patients with IgG4-related disease in our hospital.”

Q10. Kindly enumerate the limitations of the study.

A10. In consideration of some of the questions and remarks provided by the Reviewer, we have modified the limitations of the study as follows (page 19, line 298 to page 20 line 306): “This study has several limitations inherent to its retrospective design. First, the number of cases included in the analysis, especially that of IgG4-SC alone (n = 2), was not sufficient, which made it impossible to obtain statistically significant differences in some analyses including the IgG4-SC cases alone. Second, considering both the decrease in hypoechoic nodules after the steroid treatment and the previously reported findings, we indirectly demonstrated that the hypoechoic nodules were caused by inflammation triggered by IgG4-positive cell infiltration; however, we could not establish a direct relationship between pathological and US findings. In future studies, needle biopsy or cytology testing should be performed to exclude other diseases.”

Q11. A prospective study design would be worthwhile before recommending that “US findings of multiple hypoechoic lesions in SGs, which can be diagnosed non-invasively and objectively, should be widely used for the diagnosis of IgG4-related pancreato-biliary diseases.”

A11. We thank the Reviewer for their salient comment. To avoid misleading the readers, we have revised the text as follows (page 17, lines 249–252): “Thus, noninvasive and objective diagnosis of multiple hypoechoic lesions in SGs using US might have wide utility for the diagnosis of IgG4-related pancreatobiliary diseases, which warrants future prospective studies in sialadenitis and IgG4-related diseases.”

Q12. According to the study results, there is still 15-20 % risk of missing the diagnosis of IgG4-RD by using SG US, which seems significant, and the results of the study need to be confirmed with a bigger sample size before it can be uniformly considered for the diagnosis of IgG4-RD.

A12. Although we agree that invasive methods must be used for the definitive diagnosis of IgG4-related diseases, in this study we emphasize that we were able to increase the diagnostic performance for this condition by 13% without the use of invasive testing, as shown in Figure 2c.

We have revised the sentence to emphasize the significance of this study as follows (page 19, lines 295–297): “Second, the use of the ultrasonographic findings of multiple hypoechoic nodules in SGs reduced the number of cases requiring invasive testing by 13%, as shown in Fig. 2c, while diagnosing IgG4-related pancreatobiliary disease.”

Round 2

Reviewer 2 Report

1)      Why only 35 and 8 patients of AIP and IgG4-SC (out of 69) were evaluated with SG US? Why rest of the patients were not considered for SG US?

Author Response

Responses to the Comments from Reviewer #2

We would like to thank Reviewer #2 for their critical comments and useful suggestions, which have greatly enhanced the quality of our manuscript. As indicated in the point-by-point responses below, we have addressed all comments and have incorporated the corresponding changes in the revised manuscript.

Comments to the Author from Reviewer #2 (Round 2)

Q: Why only 35 and 8 patients of AIP and IgG4-SC (out of 69) were evaluated with SG US? Why rest of the patients were not considered for SG US?

A: We apologize for the confusion and inadequate revise in the previous round. We performed SG US at least once in all patients, and among them, we evaluated 35 and 8 patients with AIP and IgG4-SC, respectively, using SG US both before and after steroid therapy.

We have revised the sentence by adding “both” to avoid confusion as follows (page 6, lines 112–114): “All patients were evaluated by SG US at least once, and among them, thirty-five and eight patients with AIP and IgG4-SC, respectively, were evaluated by SG US both before and after steroid therapy.”